# Design Simulation and Data Analysis of an Optical Spectrometer

**Muddasir Naeem** [1],*[ID]**, Tayyab Imran** [1][ID]**, Mukhtar Hussain** [2][ID] **and Arshad Saleem Bhatti** [3]

1   Group of Laser Development (GoLD), Department of Physics, Syed Babar Ali School of Science and Engineering, Lahore University of Management Sciences (LUMS), Lahore 54792, Pakistan
2   Extreme Light Laboratory, Department of Physics and Astronomy, University of Nebraska-Lincoln, Lincoln, NE 68588, USA
3   M.A. Jinnah Campus, Virtual University, Defence Road, Off Raiwind Road, Lahore 54660, Pakistan
*   Correspondence: muddasirnaeem98@gmail.com

**Abstract:** Spectrometers have a wide range of applications ranging from optical to non-optical spectroscopy. The need for compact, portable, and user-friendly spectrometers has been a focus of attention from small laboratories to the industrial scale. Here, the Czerny Turner configuration-based optical spectrometer simulation design was carried out using ZEMAX OpticStudio. A compact and low-cost optical spectrometer in the visible range was developed by using diffraction grating as a dispersive element and a USB-type webcam CCD (charge-coupled device) as a detector instead of an expensive commercial diffraction grating and detector. Using National Instruments LabVIEW, data acquisition, processing, and display techniques were made possible. We employed different virtual images in LabVIEW programs to collect the pixel-to-pixel information and wavelength-intensity information from the image captured using the webcam CCD. Finally, we demonstrated that the OpticStudio-based spectrometer and experimental measurements with the developed spectrometer were in good agreement.

**Keywords:** optical spectrometer; NI LabVIEW; virtual image (VI); ZEMAX OpticStudio



## 1. Introduction

Spectrometers are used to measure light of different wavelengths over a wide range of the electromagnetic spectrum. They are widely used to analyze material by spectroscopy. C. R. Masson developed a low-cost acoustic optical spectrometer for millimeter-wave observation with an effective resolution of 160 kHz [1]. Arzhantsev, S. and Maroncelli developed and characterized a spectrometer based on an optical Kerr shutter for emission gating and a polychromatic plus charge-coupled device (CCD) detection system for recording the time-resolved emission spectra of fluorescence species [2]. An acoustic-optical spectrometer was designed and tested for solar radio astronomy and variability studies of cosmic maser sources with a 5 m antenna (RT5) [3]. An imaging spectrometer based on the curved prism configuration has been modeled, showing the imaging quality factors influencing [4]. The imaging spectrometer is an optical instrument that can simultaneously measure spectral and spatial properties. Because of its dimensional reliability and low cost, the prism-based dispersive spectrometer is one of the most used techniques in remote sensing. A micro-optical spectrometer in the spectral range of 200–910 nm was developed with a resolution of ~1 nm [5]. To improve the spectral resolution, an Offner spectrometer based on the geometrical study of ring fields was developed, and the analytical architecture was demonstrated using the optical design program Code V to construct a spectrometer with a range of 900 to 1700 nm [6]. The concept and design of an integrated optical device featuring evanescent field sensing and spectrometric analysis were presented by Gloria Mico [7]. Such an integrated optics sensing spectrometer (IOSS) consists of a modified arrayed waveguide grating (AWG) whose arms are engineered into

two sets having different focal points. Two reference designs have been provided for the visible and near-infrared wavelengths, aimed at the determination of the concentration of known solutes through absorption spectroscopy. Recently, we designed the Czerny–Turner Configuration-Based Raman Spectrometer by using Zemax based on the physical optics propagation algorithm [8]; however, the experimental validation of such a spectrometer was required for implementation.

Here, we present the design, simulation and development of a compact and low-cost optical spectrometer with data acquisition and analysis using NI LabVIEW. The simulations were carried out using Zemax OpticStudio. The spectrometer results were recorded in the form of images and then analyzed using different tools in NI LabVIEW and compared with the Zemax OpticStudio results.

## 2. General View of an Optical Spectrometer

The Czerny–Turner configuration is one of the compact and flexible spectrometer designs, consisting of a single detector instead of a detector array and requiring fixed components, as shown in Figure 1a. The Czerny–Turner Spectrometer consists of one plane diffraction grating and two concave mirrors. The first mirror is a collimating mirror that collimates the beam and makes it equivalent to the grating surface of the spectrometer. The following mirror is the focusing mirror that focuses the input light from the source on the image sensor.

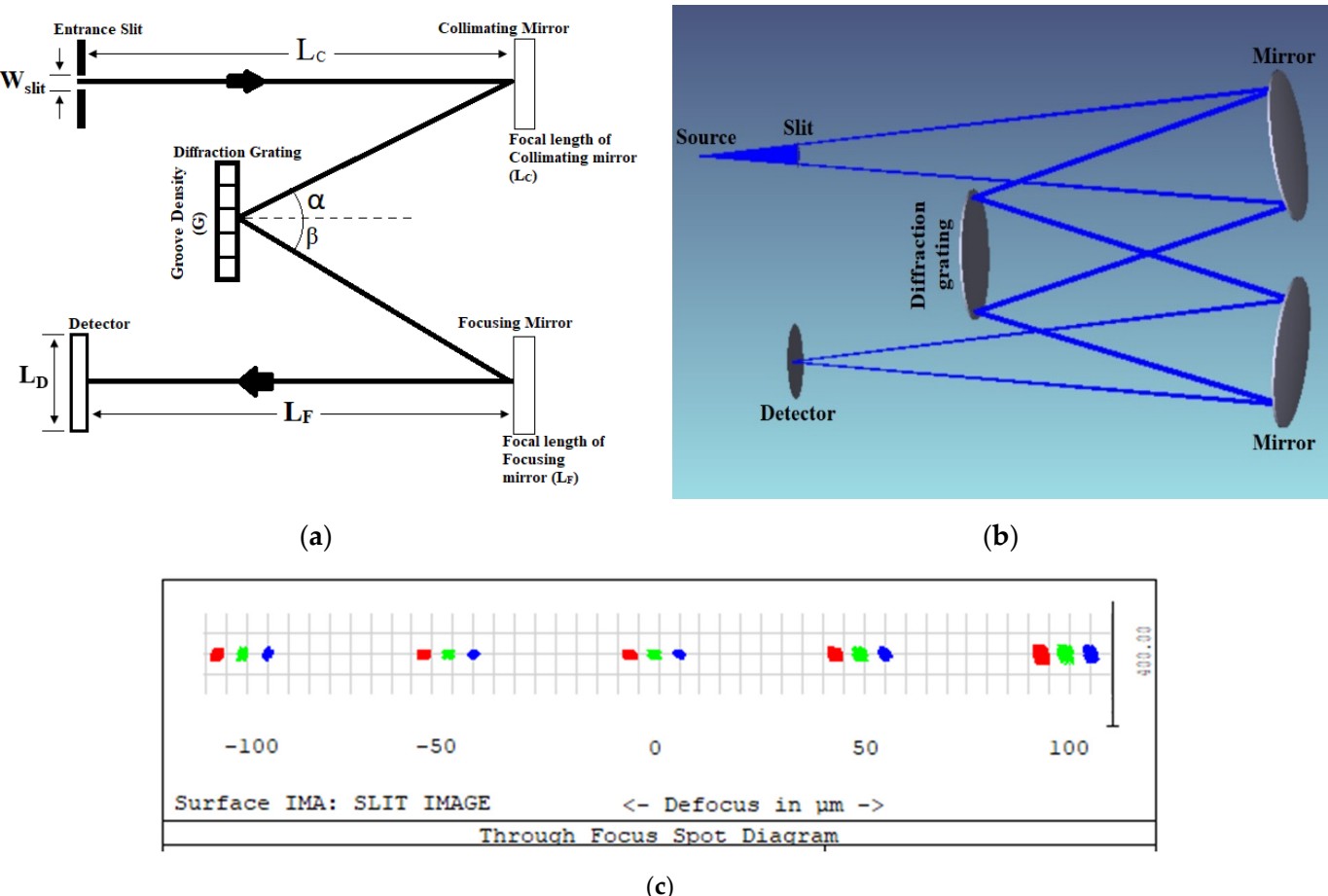

(a) (b)

(c)

**Figure 1.** (a) Schematic representation of Czerny–Turner configuration-based spectrometer; (b) schematic of the simulated design of the spectrometer in OpticStudio. (c) Variation in spot diagram relative to the distance of the detector from the focal point of focusing mirror.

Initially, the wavelength range is selected, and then the center wavelength is calculated as follows:

$$\lambda_2 - \lambda_1 = \Delta\lambda \quad ; \quad \lambda_c = \frac{(\lambda_2 - \lambda_1)}{2}, \tag{1}$$

If the deviation angle ($\theta$) = 0, then it will be the Littrow configuration [9]. According to the deviation, angles $\alpha$ and $\beta$ are calculated.

$$\alpha = \sin^{-1}\left[\frac{\lambda_c G}{2}cos\left(\frac{\theta}{2}\right)\right] - \frac{\theta}{2} \tag{2}$$

$$\beta = \theta - \alpha$$

Then, one must choose the detector size according to the size of the spectrometer. After choosing detector size, the focal length of the focusing mirror $L_F$ is obtained as,

$$L_F = L_D \frac{\cos(\beta)}{G(\lambda_2 - \lambda_1)} \tag{3}$$

The focal length of collimating mirror $Lc$ is calculated as,

$$L_c = L_F\left[\frac{\cos(\alpha)}{\cos(\beta)}\right] \tag{4}$$

By using the collimating mirror $L_C$, the slit width is obtained as,

$$W_{slit} = G(\Delta\lambda)\frac{L_c}{\cos(\alpha)} \tag{5}$$

Here, $G$ is the grating constant, and $W_{slit}$ is the slit width, which is the entrance slit for any optical light. The careful selection of the slit width is an important factor when designing and developing a spectrometer [10].

## 3. Design and Simulation of Optical Spectrometer

The design and simulations of the spectrometer were carried out using the ZEMAX OpticStudio [11] software. The parameters of each optical component defined in the lens data editor window were calculated as per the mathematical model described in the previous section and are listed in Table 1.

**Table 1.** Lens data editor window for the design of an optical spectrometer in OpticStudio.

|     | Surf: Type | Comment | Radius [cm] | Thickness [cm] | Semi-Diameter [cm] | Lines/µm |
| --- | --- | --- | --- | --- | --- | --- |
| OBJ | Standard | Source | Infinity | 0.90 | 0.00 | |
| 1 | Standard | Entrance slit | Infinity | 4.55 | 0.08 | |
| 2 | Standard | Collimating mirror | −10 | −3.00 | 0.61 | |
| 3 | Diffraction Grating | Diffraction grating | Infinity | 3.00 | 0.52 | 0.600 |
| 4 | Standard | Focusing mirror | −10 | −4.55 | 0.61 | |
| IMG | Standard | Detector | Infinity | - | 0.30 | |

We set the surface type to standard except for the diffraction grating surface. Values of parameters present in the lens data editor (LDE) were according to the designed spectrometer. After defining the required surface and other relevant parameters involved in the design, the software provided a pictorial layout of the spectrometer, as shown in Figure 1b. This included the source, input slit, collimating mirror, diffraction grating, focusing mirror and image at the last surface (detector).

The design and simulation of the spectrometer were carried out using the physical optics propagation (POP) algorithm for three different wavelengths: 0.5 µm, 0.6 µm

and 0.7 µm. The spectrometer simulation was carried out using a diffraction grating of 600 lines/mm. For the designed spectrometer image-space NA, object space NA, image-space F/#, total tracks, and stop radius, the values were 0.094, 0.088, 39.0625, 4.5 and 0.08 mm, respectively.

## 4. Simulation Results of the Spectrometer (Visible Range)

### 4.1. Spot Diagram

The variation in the size of the spot by changing the position of the detector is shown in Figure 1c. A focused spot will be obtained if the detector is placed at the mirror's focal point.

When we move the detector towards or away from the focusing mirror, the spot size on the detector varies, as shown in Figure 1c. The target's geometrical representation becomes a finite blur spot in a decreased resolution when the detector is not in the paraxial region. As the separation between these planes increases, the blur spot becomes more prominent, and the resolution decreases further.

### 4.2. Image of the Spectral Irradiance

Spectral irradiance is the irradiance of the surface per unit frequency or wavelength. The spot size at 0.5 µm, 0.6 µm and 0.7 µm, respectively, in terms of total irradiance at the surface of the detector, is shown in Figure 2. Irradiance at the center was maximum because most of the rays were focused at the center; moving away from the center decreased the irradiance and energy per unit area delivered to the surface also decreased [12,13]. The images of spectral irradiance showed that the aberration was minimum in the optical range.

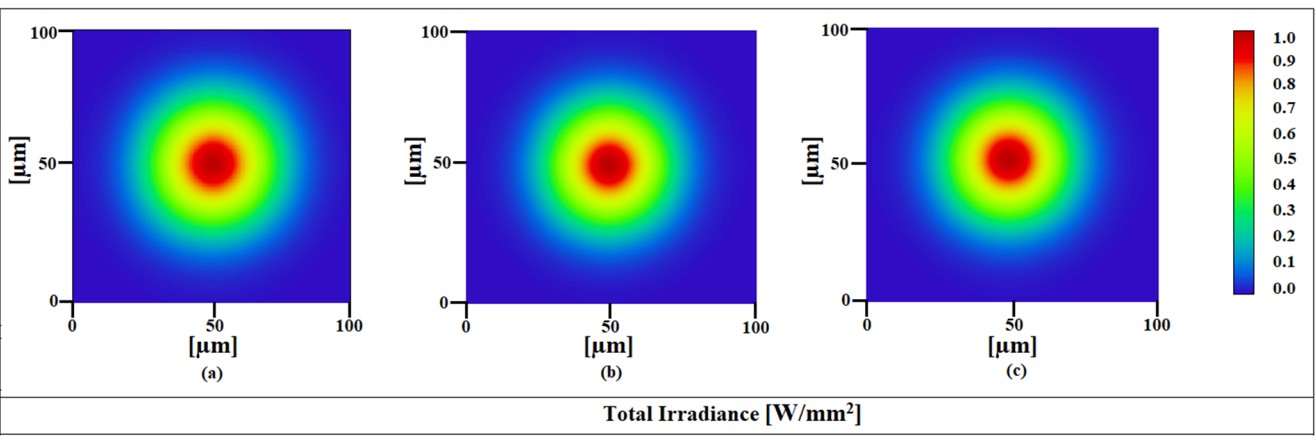

**Figure 2.** Images of the spectral irradiance on the detector at (**a**) $\lambda = 0.5$ µm (**b**) $\lambda = 0.6$ µm (**c**) $\lambda = 0.7$ µm.

## 5. Experimental Setup and Results

The schematic layout of the spectrometer setup, as shown in Figure 3a, consists of two concave mirrors, diffraction grating, white light LED as a source, and a webcam without a lens as a CCD detector. The radius of curvature of the mirrors is 10 cm, and a diffraction grating of grating constant of 600 lines/mm is used. The USB webcam CCD is used as a detector to acquire the image through NI LabVIEW. NI LabVIEW is used to record the results in the form of an image and further analyze the results using different tools in LabVIEW [14,15]. NI LabVIEW presents an easy and user-friendly interface through which we can process data by using different virtual images (VIs). The resolution of the designed spectrometer is 0.039 µm for 380 nm wavelength and 0.080 µm for 770 nm wavelength.

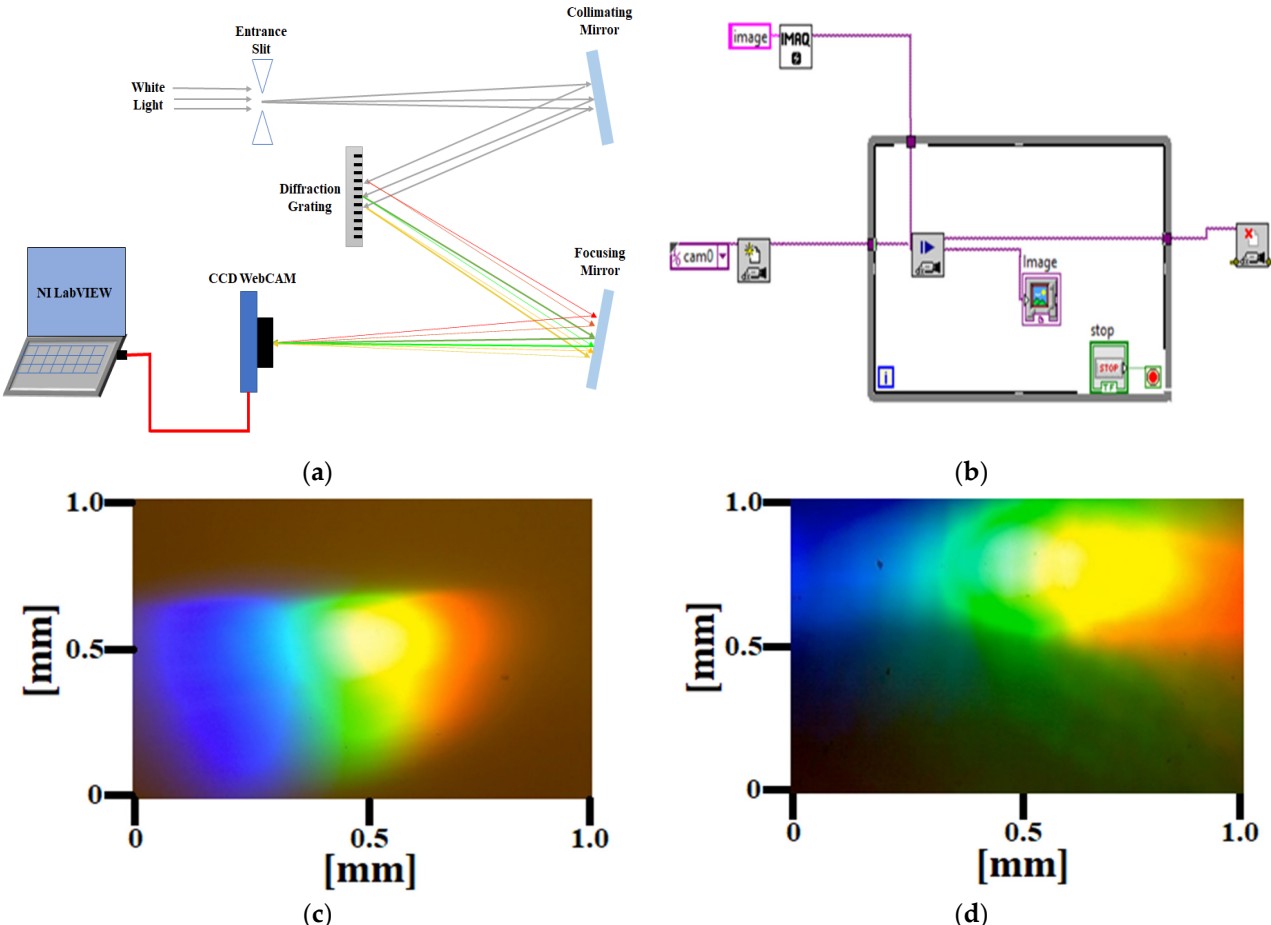

**Figure 3.** (**a**) Schematic of the experimental setup of the Czerny–Turner configuration-based optical spectrometer. (**b**) Block diagram in LabVIEW for image acquisition in snaps. (**c**) Image of spectrum acquired using NI LabVIEW program for a prism as dispersive element. (**d**) Image of spectrum acquired using NI LabVIEW program for diffraction grating as a dispersive element.

*Acquisition of Image*

The image of the output spectrum was acquired using NI-IMAQ tools in LabVIEW. Three methods for acquiring images in the NI-IMAQ tool could be adopted: snap, grab, and sequence modes. The snap method takes a single image for each retrieval, while the grab method captures the number of images, but only one image is taken for processing. Using the sequence method, only particular images could be acquired and further processed [16]. The easiest mode for image acquisition was a snap acquisition, which was considered in our acquisition, which took only one image for processing [17].

The block diagram in LabVIEW for the image acquisition in snap mode is shown in Figure 3b, which consists of IMAQ Snap, IMAQ Create, Image screen and IMAQ CloseCamera, which was used to acquire a snap through the camera. Using a while loop with these VIs, we continuously acquired the images. The snap method for image acquisition begins with the initialization stage. First, a session-in VI is used, which instructs the camera to open. We were using a USB webcam, so cam0 was identified as the default. IMAQ Create VI created a temporary memory location for an image. IMAQ Snap VI took the snap when the camera was open [16]. The image appeared on the image screen VI. In the end, IMAQ CloseCamera VI was placed, which closed the camera after taking the required snap. With the IMAQ Snap VI in the while loop, continuous snaps were recorded until the while loop was stopped using the STOP button [17].

We alternatively tested our imaging spectrometer by using prism and diffraction grating. Images recorded from the optical spectrometer using prism and diffraction grat-

ing are shown in Figure 3c,d respectively. The images obtained from the grating-based spectrometer were more dispersive than those from the prism-based spectrometer.

## 6. Analysis of Experimental Results Using LabVIEW

### 6.1. Formation of Array with Pixel-to-Pixel Values and Corresponding Graph

In the initial phase, we acquired the image using the program in NI LabVIEW. Then, different VIs were used to process the acquired image and obtain pixel-to-pixel values. For this, a program was designed using different VIs to produce a graph that displayed the intensity of each pixel. The designed program could pick pixel values from the given image and form an array of pixels [18,19]. To acquire the pixels in graphical form, we designed a program using VIs in NI LabVIEW, which yielded a graph (spectrum) of the pixels. We used a VI of IMAQ Create, IMAQ ReadFile, Invoke Node, IMAQ LineProfile and IMAQ Dispose. We used a while loop, case structure, an array, and mathematical and logical operations in this VI.

The block diagram of the program for acquisition of the graph from pixels is shown in Figure 4a. First, IMAQ ReadFile VI took the image from the address given in the file path. IMAQ Create VI created a temporary memory file of the image to process it further. Then, Invoke Node VI defined the event (draw line) and gave coordinates of the line (region of interest) drawn on the image screen. In the case structure, IMAQ LineProfile was used to calculate the profile of a line of pixels. This VI returned a data type (cluster) compatible with the LabVIEW graph. The relevant pixel information was taken from the specified line. The IMAQ LineProfile VI was used to obtain the values of pixels at the given coordinates in graphical form. The IMAQ LineProfile VI only processed the grayscale images, so first, it converted the input image into the grayscale image and then returned the values of pixels at the coordinates selected in the image. IMAQ Dispose and Error Handler VI were placed outside the while loop to remove the extra coordinates that were not running in the loop. A Nor gate condition was applied to the case structure; if it was true, then the case structure would produce output; otherwise, it would stop.

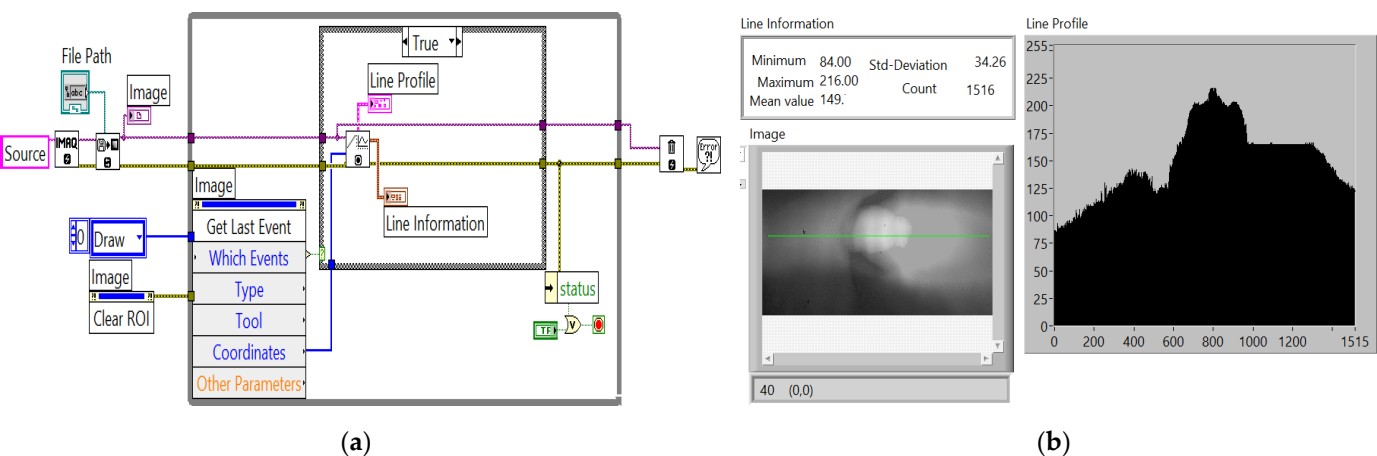

(**a**) (**b**)

**Figure 4.** (**a**) Block diagram for the line information on pixels. (**b**) Green line is drawn on the image to observe the line graph.

The green line drawn on the image screen showed that pixels were collected from the part of the image covered by the green line. The line graph of pixels is shown in Figure 4b, which shows the variation of pixels on the image. The small window on the upper left of Figure 4b gives information about the maximum pixels, minimum pixels, and deviation in pixel values in the graph.

### 6.2. Color Spectrum of RGB (Color) Image

We designed a program using different VIs in NI LabVIEW, which produced a wavelength intensity graph of the image. The block diagram to record the spectrum from the RGB image is shown in Figure 5a. The designed program used different VIs, such as IMAQ Create, IMAQ ReadFile, Invoke Node, IMAQ ColorLearn and IMAQ Dispose. We also used a for loop, while loop, case structure, arrays, and mathematical and logical operations. First, IMAQ ReadFile VI read an image file from the address given in the file path. The file format could be a standard format such as BMP, TIFF, JPEG, JPEG2000, PNG, etc., or a nonstandard format known to the user. File-path was the complete pathname, including drive, directory, and filename, of the file to read. After that, IMAQ Create VI specified the image type and name and produced a temporary memory file of the image to process it further. Then, Invoke Node VI was used to define the event and invoke a method or action on an ROI. In the first case structure, IMAQ ColorLearn VI was used to extract the color features of an image. This could be used for color matching or other applications related to color information, such as color identification and color image segmentation.

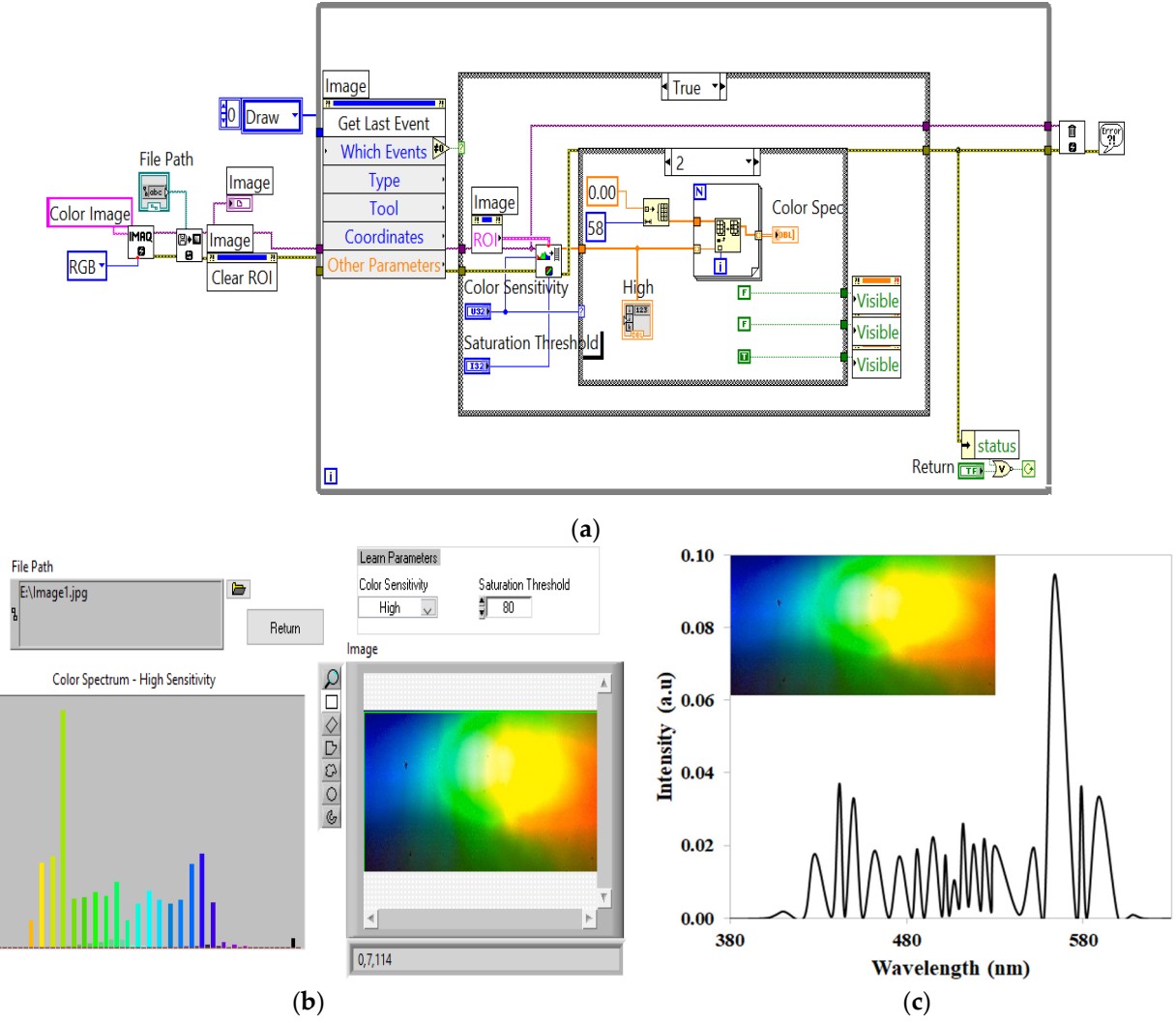

**Figure 5.** (**a**) Block diagram of the program for the color spectrum of RGB image. (**b**) Color spectrum of RGB image (front panel). (**c**) Wavelength-intensity spectrum (FWHM: in the blue region is 6.1 nm, green region is 5 nm, yellow region is 6 nm).

In ColorLearn VI, the image was a reference to the color image to learn color information; color sensitivity specified the sensitivity of the color information in the image. When

the loop ran at low sensitivity, it could collect 16 colors from the selected image. There could be 32 colors in the medium sensitivity loop, and in the high sensitivity loop, it would pick 58 colors. The learn saturation threshold specified the threshold value to distinguish two colors with the same intensity value. The color spectrum returned the color features found in the image region. These features represented the color information in the image region in a compact form [18,19].

Inside the second case structure, we used a for loop and arrays to store the intensities corresponding to the different colors. The x-axis of the spectrum contained the number of colors, and the build-in property of the VI adjusted the order of color. The y-axis gave the intensity (number of pixels) of the color in the image. Waveform graph VI was used outside the for loop to collect the data stored in the array and plot it in graphical form. IMAQ Dispose and Error Handler VI were placed outside the loop to remove the portion of the image that was not running in the loop [20,21].

The above Figure 5c shows the spectral profile of each wavelength (blue, green, yellow/reddish) acquired through the image acquisition program in NI LabVIEW. The wavelength-intensity graph was plotted by exporting the data acquired through NI LabVIEW. Spectral fringes were observed due to wavelength interference. The simulation results were compared with the experimental results, showing good agreement.

## 7. Conclusions

An optical spectrometer was designed, simulated and constructed using a diffraction grating and CCD webcam as a detector. The designed simulation of this optical imaging spectrometer was carried out in OpticStudio. Different programs were designed using NI LabVIEW tools to capture the image of the spectrum obtained at the detector. The experimental results in the form of images were converted into graphical form by using different VIs in LabVIEW. The results with the spectrometer were analyzed to obtain an array of pixels and pixel-to-pixel information on the captured image in graphical form. A LabVIEW program was used to create an intensity-wavelength graph of the results captured in the form of an image. Simulation and experimental measurements with the imaging spectrometer were in good agreement. This study highlights the potential for building low-cost, efficient imaging spectrometers for laboratories with limited sources.

**Author Contributions:** Conceptualization, T.I., M.N. and A.S.B.; methodology, M.N. and T.I.; formal analysis, M.N. and T.I.; investigation, M.N., T.I.; data curation, M.N., T.I. and M.H.; writing—original draft preparation, M.N., T.I. and M.H.; writing—review and editing, T.I., M.N. and M.H.; visualization, T.I. and M.N.; supervision, T.I. and A.S.B. All authors have read and agreed to the published version of the manuscript.

**Funding:** This research received no external funding.

**Conflicts of Interest:** The authors declare no conflict of interest.

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
