# Peer review of "Design Simulation and Data Analysis of an Optical Spectrometer"

_optics, doi:10.3390/opt3030028_

Round 1

Reviewer 1 Report

In this paper, the authors present the design, simulation and development of optical spectrometer with data acquisition and analysis using NI LabVIEW. The simulations are carried out using Zemax OpticStudio. The spectrometer results are recorded in the form of images and then analyzed using different tools in NI LabVIEW and compared with the Zemax OpticStudio results. A compact and low-cost optical spectrometer in the visible range has been developed by using diffraction grating as a dispersive element and USB-type WebCAM CCD (Charge-coupled device) as a detector instead of an expensive commercial diffraction grating and detector. The paper is novel and includes some interesting and original results, may be interested to several researchers in this field. Therefore, I recommend to accept it for publication in Optics, but with the following revisions.

1. What is the resolution of the spectrometer?

2. It works in the near infrared. What's the wavelength?

3. How stable is the spectrometer? Its ability to work continuously.

4. Is the fiber input as well as the space input okay?

Author Response

Point 1: What is the resolution of the spectrometer?

Response 1: The resolution of our designed spectrometer is 0.039µm for 380nm wavelength and 0.080µm for 770nm wavelength.

Point 2: It works in the near infrared. What's the wavelength?

Response 2: The wavelength range is 380 nm to 770 nm.

Point 3: How stable is the spectrometer? Its ability to work continuously.

Response 3: Yes the spectrometer is stable and its work continuously for long time without any problem.

Point 4: Is the fiber input as well as the space input okay?

Response 4: Yes the spectrometer is okay with both type of input.

Reviewer 2 Report

The paper is well structured and presented. 
By the way some details might be improved. A very minor point is to add a picture of the real setup, it's interesting to see the optomechanical arrangement. Other improvements could be to specify the light source used and, most important, what kind of camera is used for the experiment. In the text it's just defined as a USB Webcam CCD.
Finally, I highly suggest to implement a verification measurement performed with a commercial spectrometer (e.g. Ocean Optics or Thorlabs) as countercheck of the experimental data performed. Or to compare a real sample measurement (some chemical dyes in solution for example) both with your spectrometer and with a commercial one.

Author Response

Point 1: By the way some details might be improved. A very minor point is to add a picture of the real setup, it's interesting to see the optomechanical arrangement. Other improvements could be to specify the light source used and, most important, what kind of camera is used for the experiment. In the text it's just defined as a USB Webcam CCD.

Response 1: The picture of the experimental setup on the optical table is not a clear one to add to the article so we just add the layout diagram. We simply take an average quality Webcam and take out the CCD from it and use that CCD as our detector.

Point 2: Finally, I highly suggest implementing a verification measurement performed with a commercial spectrometer (e.g. Ocean Optics or Thorlabs) as countercheck of the experimental data performed. Or to compare a real sample measurement (some chemical dyes in solution for example) both with your spectrometer and with a commercial one.

Response 2: We compare the results of our designed spectrometer with the commercial spectrometer results, and they were in good agreement. But the article becomes too lengthy, so we remove those figures to make the article compact and clear. So, we are focusing on our designed spectrometer results in the article.
